# Dietary Live Yeast Supplementation Influence on Cow’s Milk, Teat and Bedding Microbiota in a Grass-Diet Dairy System

**DOI:** 10.3390/microorganisms11030673

**Published:** 2023-03-06

**Authors:** Isabelle Verdier-Metz, Céline Delbès, Matthieu Bouchon, Etienne Rifa, Sébastien Theil, Frédérique Chaucheyras-Durand, Eric Chevaux, Lysiane Dunière, Christophe Chassard

**Affiliations:** 1Université Clermont Auvergne, INRAE, VetAgro Sup, UMR 0545 Fromage, 20 Côte de Reyne, 15000 Aurillac, France; 2Université Clermont Auvergne, INRAE, UE 1414 Herbipôle, Domaine de la Borie, 15190 Marcenat, France; 3Lallemand SAS, 19 rue des Briquetiers, 31702 Blagnac, France; 4Université Clermont Auvergne, INRAE, UMR 0454 MEDIS, Site de Theix, 63122 Saint-Genès-Champanelle, France

**Keywords:** live yeast, dairy cow performance, milk microbiota, teat skin microbiota, bedding material microbiota

## Abstract

The supplementation of animal feed with microbial additives remains questioning for the traditional or quality label raw milk cheeses with regard to microbial transfer to milk. We evaluated the effect of dietary administration of live yeast on performance and microbiota of raw milk, teat skin, and bedding material of dairy cows. Two balanced groups of cows (21 primiparous 114 ± 24 DIM, 18 multiparous 115 ± 33 DIM) received either a concentrate supplemented with *Saccharomyces cerevisiae* CNCM I-1077 (1 × 10^10^ CFU/d) during four months (LY group) or no live yeast (C group). The microbiota in individual milk samples, teat skins, and bedding material were analysed using culture dependent techniques and high-throughput amplicon sequencing. The live yeast supplementation showed a numerical increase on body weight over the experiment and there was a tendency for higher milk yield for LY group. A sequence with 100% identity to that of the live yeast was sporadically found in fungal amplicon datasets of teat skin and bedding material but never detected in milk samples. The bedding material and teat skin from LY group presented a higher abundance of *Pichia kudriavzevii* reaching 53% (*p* < 0.05) and 10% (*p* < 0.05) respectively. A significant proportion of bacterial and fungal ASVs shared between the teat skin and the milk of the corresponding individual was highlighted.

## 1. Introduction

In the upland areas where the agricultural activities face multiple constraints (geographic, climatic, etc.), the native grasses are a well-balanced energy and protein feed for cattle and constitute the primary resource valued by the breeders [1]. Grassland-based feeding system for dairy cow production is claimed as an added value for Protected Designation of Origin (PDO) cheese quality and typicity [2]. Different strains of live yeast have been widely used as feed additives for dairy cows and as starters in many food processes [3]. In dairy ruminants, live yeasts have been shown to improve dry matter intake [4], stabilise ruminal pH [5], decrease the risk of acidosis, [6] and increase milk production [7]. Moreover, this is well-adapted and economically sustainable for a production system supporting more modest milk yields than in lowlands [8]. The addition of live yeasts in dairy cattle rations and their fate in the farm environment, potential transfer to milk, and more generally their possible influence on the microbial profile of milk, has been scarcely studied [9,10]. However, this information would be particularly relevant for traditional, quality label cheeses made from raw milk. Grilli et al. showed no impact of the distribution of live yeasts to dairy cows on the levels of cultivable microbial groups neither in milk nor in cheese [7]. Metagenomic-type approaches offer new opportunities to assess the effect of probiotics on microbiota at the individual animal level and in the farming environment. Recently, some studies supported the transfer of bacteria between cow-teat and raw milk [11,12] and another recent one hypothesized that microbial communities indigenous to raw milk may originate from permanent grazed grasslands by the intermediate of dairy cows according to the sequence soil–phylosphere–teat–milk [13], but none focused specifically on the transfers of fungal community. High-throughput amplicon sequencing of fragments of the 16S and ITS ribosomal RNA marker genes is now largely widespread to investigate the predominant and sub-dominant microbial communities and their dynamics in complex ecosystems, including the dairy chain [12,14]. Such methods are continuously and rapidly evolving, as evidenced by the numerous bioinformatic workflows developed [15,16]. Recently, the metabarcoding based on amplicon sequence variants (ASV) for microbial community characterization has become the most resolutive method to explore microbial diversity [17,18] in large-scale studies with the greatest potential to reveal intrinsic biological meaning [19].

The aim of our study was to evaluate the effect of a live yeast supplemented diet to dairy cows on their performance and on the microbiota of raw milk, teat skin, and bedding material. The composition and balance between the different microorganisms were characterized using both culture media and metabarcoding analysis. To our knowledge, this work is the first to study the impact of a long-term administration (4 months) of live yeast to a large number of dairy cows (39 individuals) on raw milk and teat skin microbiota. Its’ originality lies in the use of low-input hay diet, typical from mountainous dairy system under PDO and in the diversity and abundance analysis of both the bacterial and the fungal populations. More specifically the populations were characterized from three components (bedding material, teat skin, and raw milk) and the potential microbial transfers between teat and milk was evaluated at the individual animal level.

## 2. Materials and Methods

### 2.1. Animals

The experiment was conducted at the experimental facility of INRAE UE1414 Herbipole in Marcenat (45°15′ N, 2°55′ E; 1.135–1.215 m above sea level, Cantal, France; (https://doi.org/10.15454/1.5572318050509348E12 on 10 January 2023). Twenty-one primiparous and 18 multiparous Holstein cows (114 ± 24 and 115 ± 33 days in milk (DIM) at the beginning of the experiment) were fed with the same control diet for 2 weeks of adaptation. Then during four months, they were separated into two balanced groups, Control (C; *n* = 19) and Live Yeast (LY, *n* = 20) according to milk yield parity, days in milk, and dry matter intake measured during the adaptation period. The two groups were kept in contiguous pens in the same naturally ventilated free-stall barn facility. The rubber cattle mats of cubicles were spread once daily with a non-disinfecting solution (Saniblanc—Lhoist—Limelette—Belgium). Teats were washed with a non-detergent soap before milking and dipped with a lactic-acid based product (Filmavit II—CTH—Romans sur Isère—France) just after milking.

### 2.2. Diets

Cows were fed *ad libitum* a total mixed ration (TMR) based on grass (hay or haylage; Appendix A) including a production concentrate (Altitude group cooperative; Appendix A). The live yeast *Saccharomyces cerevisiae* CNCM I-1077 (Levucell SC TITAN, Lallemand SAS, Blagnac, France) was supplemented at the target rate of 1 × 10^10^ CFU/d and included in the production concentrate of the LY group through a clay-based premixture to guarantee an even distribution. Unsupplemented cows (C-group) were offered the concentrate including the carrier only. At the start of the supplementation, cows were averaging 129 days of lactation for each group. Every month, the main forages and the TMR were analysed by near infrared spectroscopy (Rock River Laboratory, Watertown, WI, USA) and the live yeast count of the concentrate was checked by Lallemand’s laboratory.

### 2.3. Milk Production Parameters Measurements

Milk yield was recorded automatically at the milking parlour. Milk composition (fat, protein, lactose, urea, casein, and somatic cell count) was determined by mid-infrared spectroscopy (Agrolab’s laboratory, Aurillac, France) from individual milk samples. Cows were weighed (BW) immediately after the morning milking.

### 2.4. Microbial Analyses

#### 2.4.1. Sampling Procedure

The individual milk (C = 19, LY = 20), the teat skin (C = 9, LY = 10) as well as the bedding material (*n* = 2, 1 per pen) were collected at the evening milking at three different time points: before the beginning of yeast supplementation (P1), and after three (P2) and four months (P3) of supplementation. The individual teat skin samples were collected as described by Verdier-Metz et al. [20]. After teat cleaning and elimination of the first drafts of milking and just before fitting the clusters, two hundred mL of milk was collected by hand into a sterile bottle then stored at 4 °C overnight. Sterisox wipes (Sodibox—Nevez— France) were used to sample the bedding material in the two animal pens by walking two steps per individual cubicle. Collected samples were stored at 4 °C overnight in an individual stomacher bag, blended with 175 mL of salt tryptone (1 g/L), and the suspensions were then extracted.

#### 2.4.2. Microbial Enumeration

Total mesophilic bacteria, lactic acid bacteria, Gram-negative bacteria, and ripening bacteria (Gram-positive catalase-positive bacteria) were enumerated on different culture media as described by Monsallier et al. [21] in the individual milk of the same animals that were sampled for the teat skin analysis (C = 9 and LY = 10 per period). Yeasts were enumerated on OGA medium, which is elective for a wide spectra of yeasts and molds from dairy products while also inhibiting bacteria.

#### 2.4.3. DNA Extraction

The suspensions of teat (C = 9 and LY = 10 per period) and bedding material (P1 = 1 per pen; P2 = 2 per pen; P3 = 3 per pen) were centrifuged (8000× *g*, 20 min, 4 °C) and the supernatants were removed. Milk samples (150 mL) were centrifuged (5300× *g*, 30 min, 4 °C) then the fat and the supernatant were removed. The pellets obtained were mixed with 1 mL sterile phosphate-buffered saline, centrifuged (13,000× *g*, 5 min, 4 °C) and stored at −20 °C. The total DNA extraction was performed from the pellets using a FastDNA Spin Kit for Soil (MP Biomedicals—Illkirch—France) with bead beating.

#### 2.4.4. High-Throughput Sequencing of 16S and ITS Amplicons

The 16S rRNA genes from teat suspensions and milk samples were pre- amplified using the universal bacterial primers W02 (5′-GNTACCTTGTTACGACTT-3′) and W18 (5′-AGAGTTTGATCMTGGCTCAG-3′) as described by Verdier-Metz et al. [22] for 17 cycles. The variable region V3–V4 of the 16S rRNA gene was amplified from 2 μL of pre-amplified DNA with primers MSQ-16SV3F (5′-TACGGRAGGCWGCAG-3′) [23] and PCR1R-460 (5′-TTACCAGGGTATCTAATCCT-3′) as described by Frétin et al. [12]. To target the ITS2 region, the extracted DNA from teat suspensions and milk samples were pre-amplified using the primers NL4 (5′-GGTCCGTGTTTCAAGACGG-3′) and ITS5 (5′-GGAAGTAAAAGTCGTAACAAGG-3′) as described by Irobi et al. [24] for 17 amplification cycles. Then 2 μL of pre-amplified DNA were amplified with primers ITS3f (5′-GCATCGATGAAGAACGCAGC-3′) and ITS4_KYO1 (5′-TCCTCCGCTTWTTGWTWTGC-3′) as described by Bokulich and Mills [25] for 30 cycles. All the amplicons were sequenced using Illumina MiSeq technology (INRAE, GeT-PLaGE plateform) with the 250 bp paired-end V3 chemistry. The microbial communities were characterized using the workflow rANOMALY [26] that uses amplicon sequence variants (ASV) as taxonomic units. The 16S ASVs were assigned with an environment specific database (DAIRYdb [27]) and a general database (SILVA r138) and the ITS amplicon sequences with UNITE v8.2 and UTOPIA v072019 [28] databases. The ITS rDNA sequence of the *Saccharomyces cerevisiae* CNCM I-1077 strain was compared with all sequences of ASVs assigned to the order *Saccharomycetales* using blastn.

### 2.5. Statistical Analyses

#### 2.5.1. Milk Production Parameters

Statistical analysis was performed on 34 cows in total (17 per group, after removal of 2 outliers in C and 3 outliers in LY group) using a general linear mixed model with repeated measures (weeks of supplementation). The live yeast supplementation, the week, and their interaction were used as fixed effects, and the cow as random factor (SPSS Statistics 24.0, IBM). Performance of the adaptation period was set as covariate. Energy corrected milk (ECM) was determined according to an equation adapted from the equation of Tyrrell and Reid [29].

#### 2.5.2. Microbial Data

The milk microbial population counts and the number and proportion of ASV from the 2 experimental groups were processed by ANOVA with the software R (version 3.6.3) factoring in cow group and period. The alpha-diversity (Observed and Shannon indexes) were analyzed by ANOVA and beta-diversity (Bray Curtis dissimiliraty and generalized Unifrac methods and non-metric multidimensional scale) by permutational multivariate analysis of variance (PERMANOVA Adonis). Differential analysis was performed with three methods in ExploreMetabar [30]: DeSeq2 [31], MetaGenomeSeq [32] and MetaCoder [33]. The spearman correlation coefficients and their significance levels were calculated using R package “corrplot” between the relative abundances of the bacterial and fungal species above 1% of abundance obtained from teat samples [34] and results were illustrated in a correlogram (Appendix A).

## 3. Results

### 3.1. Animal Performance Data

No significant differences were found on dry matter intake (DMI; 20.8 ± 0.3 kg) nor on body weight (BW; 651.9 ±13.5 kg) between dietary treatment groups over the whole trial. We observed a tendency for a higher milk yield in LY cows, who produced approximately 3.1% more milk (Table 1). However, the significant effect of interaction between week and supplementation highlighted a better FCR for LY group than C group (−3.1%, Figure 1a) at the end of the experimental period. It also illustrated a greater LY milk yield at the weeks 3, 5, 7 and 8 and at the weeks 15 and 16 of the trial, whereas milk yields from both groups were closer between weeks 9 and 14 (Figure 1b). From week 14 until the end of the trial, Energy Corrected Milk (ECM) yield was also greater in LY group compared to C group (Figure 1c).

### 3.2. Milk Characteristics

#### 3.2.1. Chemical Composition

A significant effect of LY supplementation on lactose (*p* < 0.01) was observed as well as a tendency for a greater milk fat yield (Table 1). No significant difference was found on somatic cell counts (SCC) between groups. A LY supplementation x week interaction was significant for all parameters (*p* < 0.01).

#### 3.2.2. Microbial Composition

The levels of total mesophilic flora and ripening bacteria in the milks of the LY group were significantly higher than in those of the C group whatever the period (Figure 2). An interaction group x time was observed for the ripening bacteria and the yeasts: their levels increased over time in LY milk while variations less than 0.5 log10 CFU/mL were observed in C milk. The Gram-negative bacteria levels, though not significantly different between groups, decreased from the period P2 to P3. Although differences were not significant, lactic acid bacteria levels were slightly higher in LY milks compared to C milks. Among the ten LY milks, eight (3 in P2 and 5 in P3) showed yeast counts greater than 0.5 log10 CFU/mL against a single C milk in P1.

### 3.3. Dispersion of Saccharomyces Cerevisiae CNCM I-1077 like DNA in the Environment

No cultivable yeast was detected in the samples collected from the Control concentrate. In the LY concentrate samples, the total yeast concentration was on average of 9.6 ± 0.2 log10 CFU/kg of pellet, which was in agreement with the expected concentration (9.7 log10 CFU/kg). Among the total ASVs assigned to *Saccharomycetales* within the total ITS amplicon dataset, a unique ASV (8a7e93f2) was assigned to the species *Saccharomyces cerevisiae*. The 381-base pair sequence of the ASV 8a7e93f2 was identical (100% sequence identity and 100% coverage) to that of the strain *Saccharomyces cerevisiae* CNCM I-1077 introduced into the diet of the LY cows. It was detected sporadically in the bedding material and in the teat suspensions as shown in Figure 3a. Never detected in those samples during the period P1, though it was present at low relative abundance between 0.0003% and 0.0012% in LY bedding material in P2 and P3 and lower than 0.0004% in C bedding material only in P3. Two C cows (1 in P2 and 1 in P3) and seven LY cows (5 in P2 and 2 in P3) carried this ASV on their teat surfaces with a relative abundance of less than 0.01%, which is slightly higher than in bedding material but remains very low (Figure 3b). This ASV was not found in any milk sample.

### 3.4. Diversity and Composition of the Fungal Communities

A total of 3,419,231 reads were distributed to 345 fungal ASVs across 167 samples after filtration steps. ANOVA did not show significant effect of the yeast supplementation on the fungal alpha-diversity neither in teat suspensions nor in milk samples (Figure 4a). It was not possible to conclude on the effect in bedding material due to the low number of samples. The milk richness (Observed) and diversity (Shannon) indices were slightly but significantly lowered (*p* < 0.001) over time, unlike those of teats which were not affected by the period. The distribution of the fungal profiles of LY milks was not significantly different from C ones as evidenced by the betadiversity indexes (Appendix A), while in contrast, yeast supplementation impacted the fungal composition of the teat surface. The teat suspension profiles clustered both according to time and according to group as shown by the non-metric multidimensional scaling (NMDS) ordination (Appendix A, Figure 4b). The fungal profiles of the bedding material markedly changed between P1 and P3 (Figure 4c).

In P1, the relative abundances of the 20 most abundant fungal genera from the bedding material were similar in the C and the LY groups. In P3, the proportions of fungal taxa were different between the two groups: the C bedding material showed a dominance of the *Galactomyces* genus while the *Pichia* genus predominated in LY bedding material as confirmed by the differential analysis (Table 2) with a relative abundance of 44.06% (*p* < 0.1) and 55.56% (*p* < 0.05) respectively.

The differential analysis of fungal taxa between the C and LY samples (Table 2) revealed that the P3 bedding materials differed by 12 taxa at the genus level, 7 at the species level, and 5 at the ASV level. At P2 and P3, teat suspensions and milk samples were distinguished by 3 and 2 genera, 7 and 2 species, and 4 and 1 ASVs, respectively. *Pichia kudriavzevii* was the dominant fungal species and ASV both in the LY bedding material and teat suspensions with an abundance of over 53% (*p* < 0.05) and 10% (*p* < 0.05) respectively. Likewise, the genus *Aspergillus* presented higher abundance both in LY milk (5.87%; *p* < 0.05) and bedding material (1.76%; *p* < 0.05). In the C group, the most abundant identified taxa were assigned to the genus *Saturnispora* in the bedding material (13.44%) regardless of identification level, to *Wickerhamomyces anomalus* in the teat suspensions (21.27%; *p* < 0.05) and to *Pichia fermentans* in the milk samples (1.59%, *p* < 0.05).

### 3.5. Shared Fungal ASVs between Teat and Milk

The total number of ASVs identified in the three components compartments were similar (Figure 5a) from one group to another (299 in C group versus 303 in LY group).

Overall, the C and LY milk samples shared 70.5% and 72.9% respectively of their ASVs with teat suspensions, and 46% and 52.3% respectively with bedding material. The ASV 8a7e93f2, with an identical sequence to that of *Saccharomyces cerevisiae* CNCM I-1077 strain, was exclusively shared by the bedding material and the teat suspensions. Individual milk samples counted an average of 8 to 18 fungal ASVs of which 33 to 65% were shared with the teats. Neither the number of milk ASVs shared with the teats nor their cumulative abundance were significantly different between the two groups of cows (Table 3) but both increased slightly over time (*p* < 0.1).

### 3.6. Diversity and Composition of the Bacterial Communities

After quality filter sequences, a total of 2,390,892 sequence reads were obtained from all the samples and distributed into 521 bacterial ASVs. Whether in milk samples, teat suspensions, or bedding materials, live yeast supplementation had no effect neither on the alpha-diversity (Appendix A) nor on the beta-diversity (Appendix A) of the bacterial profiles. No bacteria were differentially abundant between C and LY bedding material. Two bacterial genera and three species were differentially abundant between C and LY milks (Table 2): The genus *Enterococcus* and the species *Acinetobacter johnsonii* were more abundant in the LY milks than in the C ones with a respective abundance of 5.34% and 1.83%. Conversely, the genus *Vibrionimonas* and the species *Pseudomonas alcaliphila* and *Vibrionimonas magnilacihabitans* were more abundant in C than LY in milks. While the C teat suspensions were distinguished by a higher abundance of the genus *Jeotgalicoccus*, the LY teat suspensions differed by containing a greater abundance of the genus assigned to *Staphylococcus*; of which, 16 ASVs were differentially abundant between the two groups: 4 belonged to the species *S. hominis*, 6 to the species *S. chromogenes*, 5 to the species *S. haemolyticus*, and 1 to the species *S. pseudintermedius*. No sequence was assigned to *Staphylococcus aureus*, neither on teats nor in milk samples or bedding materials. Only two ASVs assigned to *Aerococcus suis* and *Staphylococcus hominis* were more abundant both in LY milk samples and in LY teat suspensions compared to C reservoirs.

### 3.7. Shared Bacterial ASVs between Teat and Milk

No effect of group was observed neither on the number of bacterial ASVs in milk and teats nor on that of the ASVs shared between these two components (Figure 5b) and their abundance. On the other hand, the period had a highly significant effect on these criteria which showed a decrease over time (Table 3).

### 3.8. Correlation between Bacterial and Fungal Dominant Species in Teat Microbiota

Forty-seven significant correlations (24 positive and 23 negative, *p* < 0.05) between bacterial and fungal species abundances were found in the C teat suspensions versus 31 (11 positive and 20 negative) in the LY group. Of these, 20 correspond to the same bacterial and fungal species pairs, and are of the same correlation for C and LY teats with the exception of *Aerococcus suis* and *Entomophtoromycota* sp. which are positively correlated with C and negatively with LY. Non-fermentans *Pichia* species were correlated with bacterial taxa as *Intestinibacter bartlettii*, *Romboutsia* species, and *Peptostreptococcaceae* negatively in C (*p* < 0.05).

## 4. Discussion

*S. cerevisiae* CNCM I-1077 supplementation over the 4 months of experiment induced an increase in animal productivity within the usual range expected by dairy cows managed in mountainous areas [8]. Although the DMI and BW were similar between groups, the FCR was improved as well as the milk yield and ECM. In a meta-analysis, dairy cows supplemented with the same live *S. cerevisiae* yeast strain has been shown to increase feed efficiency and milk yield without impact on DMI [35]. It has also been reported that administration of this yeast strain increased the colonization of fiber by ruminal populations, notably cellulolytic bacteria and fungi [36], thus enhancing fiber digestibility and feed efficiency [35,37], and ultimately milk yield. Many publications report the crucial role that rumen microbiota plays on animal performance, in particular milk production [38,39]. Ruminal microbial populations were not studied in this experiment, but the expected outcomes in milk production were noted and confirmed the LY effect.

To our knowledge, this study was the first to investigate the impact of live yeast supplementation on both bacterial and fungal populations in dairy cow environments. More specifically, the bedding areas and the teat skin of the animals considered as a major reservoir of microbial diversity in milk were investigated as part of the environmental factors [40]. The levels of the major microbial groups evaluated by culture methods remain a relevant indicator for milk producers. Yeast levels in the individual milk samples were found to be very low compared to the literature [20,41] and highly variable from one cow to another. In the present study, the individual milk samples were collected by hand using gloves, i.e., without going through the milking machine whose internal biofilm is deemed to be one of the main recontamination sources of milk [42]. A slightly higher yeast level in some individual milks from the LY group was noticed. However, the high-throughput sequencing of amplicons did not reveal, in any individual LY milk, any ASV similar to the *Saccharomyces cerevisiae* CNCM I-1077 strain introduced into the LY diet. Although, that ASV 8a7e93f2 was sporadically identified in the bedding material and the teat suspensions. No transfer of the introduced live yeast from farm environments into the milk was thus observed.

However, live yeast supplementation had a significant impact on fungal community profiles in the bedding areas of dairy cows and on their teat skin with a major increase in the abundance of *Pichia kudriavzevii* in these two environments. In their work on the identification and characterization of yeasts from bovine rumen, Fernandes et al. [43] evidenced that *Pichia kudriavzevii* was the predominant yeast present in ruminal fluid from the cattle herds. It is therefore not surprising to find this yeast in the bedding areas, probably via the dung, and therefore also on the teat surface. Recently, Lamarche et al. [44] confirmed the presence of this indigenous fungus in the raw milk and some artisanal cheese cores. Moreover *Pichia kudriavzevii* has been suggested as being beneficial for cow performance as supplementation with this live yeast originally isolated from the rumen fluid of high-yielding dairy cows had significantly increased milk yield when supplemented to low-yielding cows [45]. *Mucor racemosus,* detected in higher abundance on the LY teat, has been previously isolated from an industrial camembert cheese during the ripening [46] and is considered as a cheese spoiler in some cheese types. Conversely, *Wickerhamomyces anomalus* was found in higher abundance on the C teats. This fungus known to originate from the diet was found in the dairy cow rumen albeit with no known role in rumen feed degradation [47]. While fungi in dairy products have been the subject of various studies [48,49], the literature on fungal microorganisms in farm environments is relatively sparse. In our study, the LY teat skin presented a higher abundance in *Candida railenensis* and *Bettsia alvei*, two fungi never identified neither in dairy products nor in dairy farm environments.

With regard to the bacterial population levels, the total mesophilic bacterial count did not exceed 2.7 log(CFU/mL) which was quite low but already observed in a previous study carried out on the same experimental farm [20]. In an attempt to obtain individual data in this study, milk samples were harvested manually, which may have also contributed to such levels. The impact of the live yeast supplementation on the bacterial community affected fewer genera, species and ASVs than on the fungal community in the milk samples and teat suspensions. None of the *Staphylococcus* ASVs identified at present were assigned to pathogenic species. The LY supplementation was associated with increased abundance of the *Staphylococcus* genus both in the milk and on the teat surface, but only of coagulase-negative staphylococci such as *S. hominis*, *S. chromogenes* and *S. haemolyticus,* which were identified as the predominant species in milk, teat apices, and rectal feces of dairy cows [50]. Moreover, no mastitis event (subclinical or clinical) was detected over the experiment and lower SCC were observed in LY group compared to C one. The *Enterococcus* genus that was identified significantly more abundant in LY milk than in C milk is widely distributed in raw-milk cheeses and is generally thought to positively affect flavor development [51]. Likewise, an ASV assigned to *Aerococcus suis* was more abundant in the LY milks and teat suspensions compared to the C components; interestingly, *Aerococcus* spp. appeared to be a promising genus as a source of bioprotective agent against *Salmonella* in dairy matrices [52].

While the bacterial microbiota of the teat skin has been the subject of several publications [12,53], the fungal populations have been very rarely studied. Consequently, the study of correlations between these two populations is inexistent to our knowledge. For the first time on the surface of cow teats, we were able to identify negative and positive correlations between bacterial and fungal phylotypes. A study involving a larger number of teat skin samples would consolidate the bacteria-fungi correlations observed in our study. However, we highlighted that some correlations patterns differed between C and LY teats and that they involved *Pichia* sp. and *Aerococcus suis*, which were more abundant on LY teats.

## 5. Conclusions

The dietary microbial supplementation with live yeast *Saccharomyces cerevisiae* CNCM I-1077 over four months improved the feed conversion ratio, milk yield, and the ECM of dairy cows in a semi-mountain dairy farming system where the animals were fed a ration based on grass (hay or haylage). Using a molecular tracking approach, the probiotic dissemination was monitored and demonstrated the absence of transfer of the probiotic yeast to the milk. Supplementation with live yeast did not have a deleterious effect on the composition of milk microbiota, which contributes to the differentiation and terroir identity of PDO cheeses. Interesting changes in the microbial balance of three different components in the dairy cow environmental were revealed in particular by the individual analysis at the ASV level such as the higher abundance of *Pichia kudriavzevii* in the bedding material and on the teat skin from LY group. In our study, the metabarcoding derived ASVs allowed to highlight for the first time that milk shared a significant proportion of bacterial and fungal ASVs with teats at the individual level, and irrespective of live yeast supplementation.

## Figures and Tables

**Figure 1 microorganisms-11-00673-f001:**
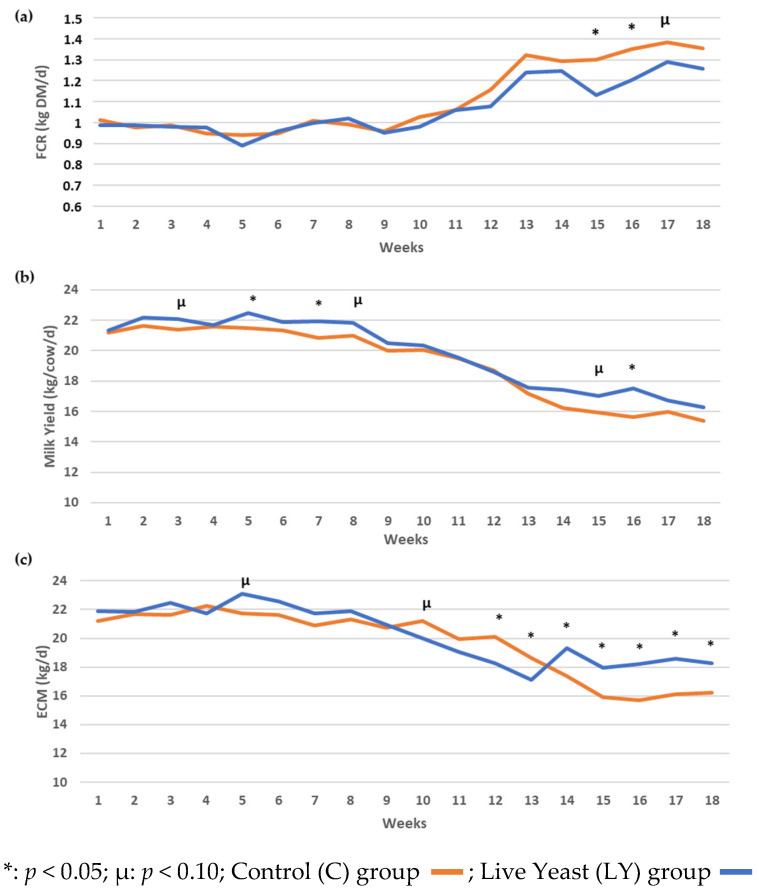
Mean of (**a**) feed conversion ration (FCR); (**b**) milk yield; (**c**) energy corrected milk (ECM) in Control (C) and Live Yeast (LY) groups over the 18 weeks of experiment.

**Figure 2 microorganisms-11-00673-f002:**
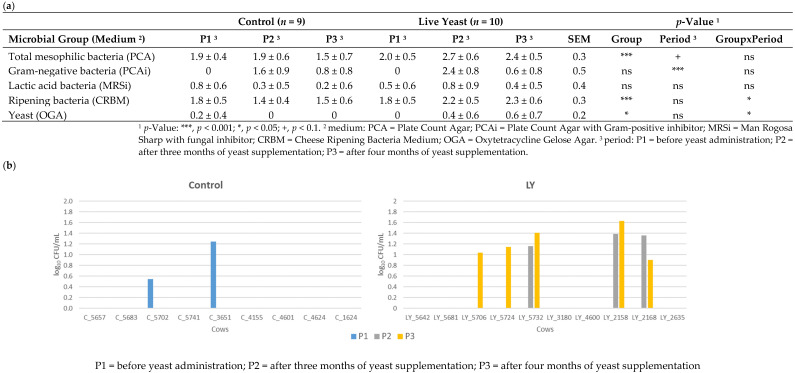
Microbial counts in individual milk samples according to the group and the period: (**a**) Mean population counts (log_10_ CFU/mL) of the main microbial groups; (**b**) Population counts (log_10_ CFU/mL) of yeast in individual milk samples as determined on OGA medium.

**Figure 3 microorganisms-11-00673-f003:**
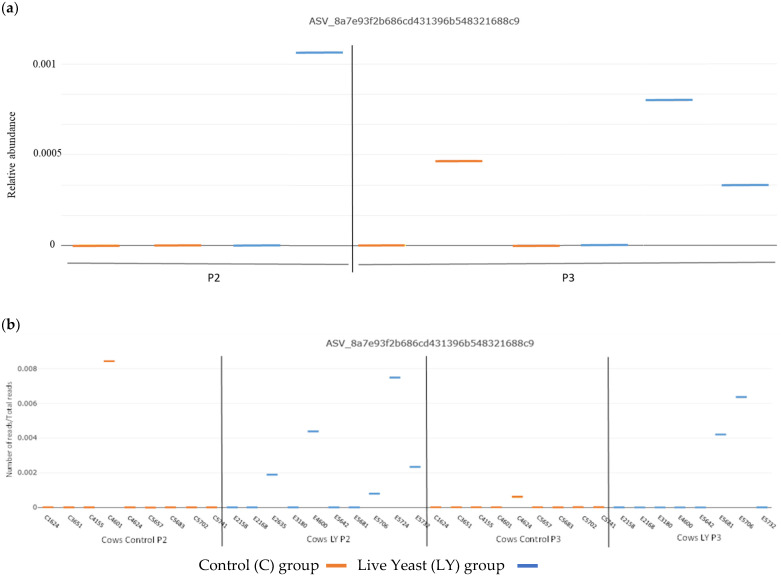
Relative abundance of the ASV with 100% of identity with *Saccharomyces cerevisiae* CNCM I-1077: (**a**) in P2 and P3 for bedding material; (**b**) in individual teat suspensions in Control (orange) and LY (blue) groups.

**Figure 4 microorganisms-11-00673-f004:**
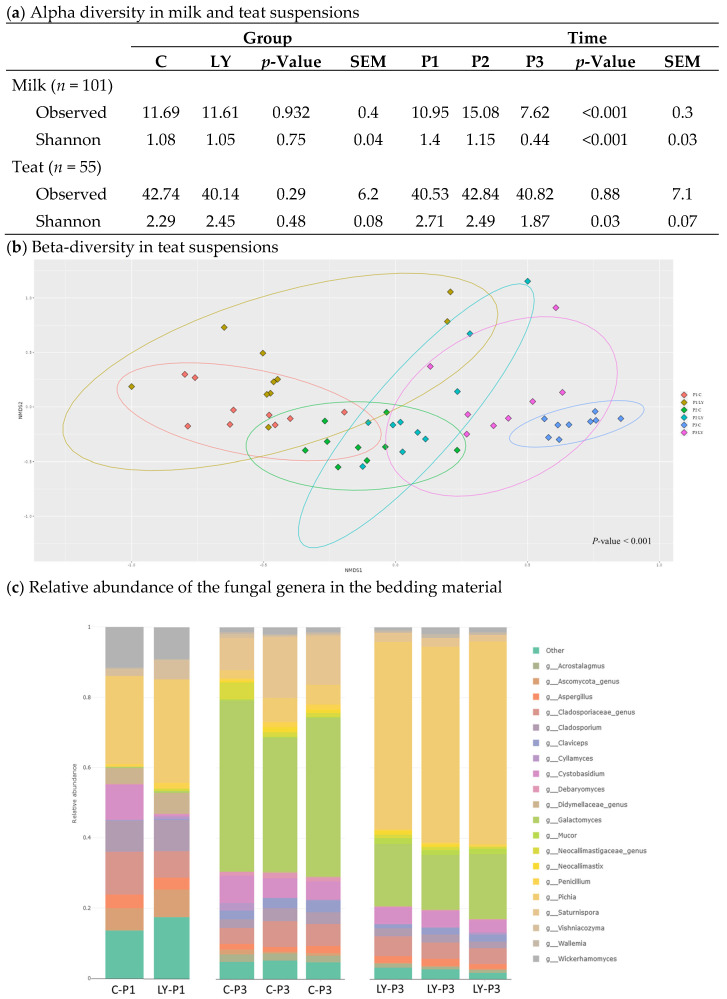
Fungal diversity according to the group (C = control or LY = live yeast supplementation) and the period (P1 = before yeast supplementation; P2 = after three months of yeast supplementation; P3 = after four months of yeast supplementation): (**a**) Alpha-diversity indexes calculated from the fungal profiles in the raw milk and the teat suspensions; (**b**) Beta diversity in teat suspensions estimated by using Bray Curtis dissimilarity method and non-metric multidimensional scale (NMDS) for which all *p*-values of all pairwises were less than 0.05; (**c**) Relative abundance of the fungal genera in the bedding material.

**Figure 5 microorganisms-11-00673-f005:**
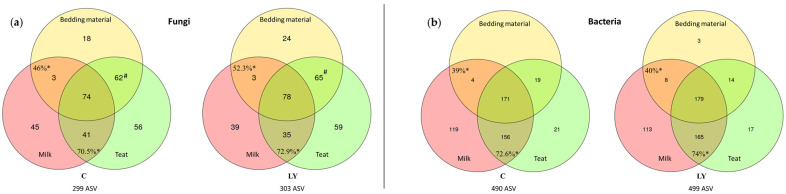
Number of: (**a**) fungal ASVs and (**b**) bacterial ASVs shared between milk, bedding material, and teat suspensions according to the group (C = control vs. LY = live yeast). *: percentage of milk ASVs shared with teat or bedding material. #: sequence of *Saccharomyces cerevisiae* CNCM I-1077.

**Table 1 microorganisms-11-00673-t001:** Live yeast supplementation (LY) effect on production performance of dairy cows.

Parameter	Control	LY	SEM	LY Effect *
Dry matter intake (DMI, kg/d)	20.62	20.99	0.30	0.397
Feed Conversion Ratio (FCR)	1.117	1.088	0.024	0.391
Milk yield (kg/d)	19.17	19.82	0.27	0.098
Energy corrected milk (ECM, kg/d)	19.69	20.28	0.35	0.246
Fat content (g/kg)	37.06	37.03	0.39	0.954
Protein content (g/kg)	30.50	30.51	0.21	0.969
Milk Fat Yield (kg/d)	0.702	0.731	0.01	0.098
Milk Protein Yield (kg/d)	0.583	0.602	0.01	0.287
Casein (mg/kg)	274.3	278.7	1.6	0.058
Lactose (g/kg)	49.5	50.8	0.2	<0.01
Urea (mg/L)	312.4	313.3	7.9	0.942
Somatic Cell Count (Log_10_/mL)	5.08	5.00	0.06	0.352
Body Weight start (BW, kg)	635.3	648.0	5.2	0.087
Body Weight end trial (BW, kg)	636.2	667.6	13.5	0.114

* live yeast effect; for all parameters, a LY x week interaction was significant, *p* < 0.01.

**Table 2 microorganisms-11-00673-t002:** Mean relative abundance (>1%) of ASVs, species, and genera showing significant differential abundance in milk samples and teat suspensions (after three (P2) and four (P3) months of supplementation) and in bedding materials (P3) between control (C) and live yeast supplemented (LY) group.

		Milk					Teat					Bedding Material			
Microbes Level	Taxa	Methods^1^	C	LY	Sign.^2^		Methods^1^	C	LY	Sign.^2^		Methods^1^	C	LY	Sign.^2^
			*n* = 38	*n* = 40				*n* = 18	*n* = 20				*n* = 3	*n* = 3	
**Fungi**															
ASV	*Candida__railenensis*	-	-	-			3	0	1.61	*		-	-	-	
	*Cystobasidium_laryngis*	-	-	-			-	-	-			1	4.84	1.01	*
	*Cystobasidium_pallidum*	-	-	-			-	-	-			2	0.17	1.47	*
	*Mucor_racemosus*	-	-	-			3	0.24	2.95	*		-	-	-	
	*Pichia_fermentans*	1	1.59	0.03	*		-	-	-			-	-	-	
	*Pichia_kudriavzevii*	-	-	-			1, 3	0.03	10.6	*		1, 2	0.02	53.09	*
	*Saturnispora*_*silvae*	-	-	-			2	2.3	0.46	*		1	8.36	1.52	*
	*Saturnispora*_species	-	-	-			-	-	-			1	2.05	0.33	*
Species	*Acrostalagmus*_species	3	1.1	0	+		3	1.83	0.34	+		-	-	-	
	*Aspergillus*_species	3	1.31	2.94	+		-	-	-			-	-	-	
	*Bettsia_alvei*	-	-	-			3	0.4	4.6	+		-	-	-	
	*Candida__railenensis*	-	-	-			3	0	1.9	*		-	-	-	
	*Claviceps_purpurea*	-	-	-			-	-	-			1	1.62	0.37	*
	*Cystobasidium_laryngis*	-	-	-			-	-	-			1	5.16	1.01	*
	*Cystobasidium_pallidum*	-	-	-			-	-	-			1	0.81	3.25	*
	*Debaryomyces_hansenii*	-	-	-			-	-	-			1	1.26	0.23	*
	*Mucor_racemosus*	-	-	-			3	0.24	3.45	*		-	-	-	
	*Neocallimastigaceae*_sp	-	-	-			-	-	-			1	2.24	0.32	*
	*Pichia_kudriavzevii*	-	-	-			1, 3	0.05	11.33	*		1, 2	0.02	53.09	*
	*Saturnispora*_species	-	-	-			2	9.45	1.1	*		1	13.44	2.31	*
	*Wickerhamomyces_anomalus*	-	-	-			1, 3	21.27	6.14	*		-	-	-	
Genus	*Acrostalagmus*	3	1.1	0	+		-	-	-			3	1.98	0.98	+
	*Aspergillus*	3	1.31	5.87	*		-	-	-			1	1.6	1.76	*
	*Cladosporium*	-	-	-			-	-	-			3	3.09	2.08	+
	*Claviceps*	-	-	-			-	-	-			3	2.95	1.67	+
	*Cystobasidium*	-	-	-			-	-	-			3	6.15	4.26	+
	*Debaryomyces*	-	-	-			-	-	-			1	1.26	0.23	*
	*Galactomyces*	-	-	-			-	-	-			3	44.06	17.15	+
	*Kurtzmaniella*	-	-	-			1, 3	0	1.9	*		-	-	-	
	*Mucor*	-	-	-			-	-	-			1, 2	0.29	1.53	*
	*Neocallimastigaceae*_genus	-	-	-			-	-	-			3	2.36	0.85	+
	*Penicillium*	-	-	-			-	-	-			3	1.16	0.51	+
	*Pichia*	-	-	-			-	-	-			1, 2	4.99	55.56	*
	*Saturnispora*	-	-	-			2	9.45	1.1	*		1	13.44	2.31	*
	*Wickerhamomyces*	-	-	-			1, 3	21.27	6.14	*		-	-	-	
Bacteria															
ASV	*Aerococcus_suis*	3	0.33	1.24	*		3	1.13	4.41	*		-	-	-	
	*Staphylococcus_hominis*	1	0.08	1.41	*		3	0.03	2.86	*		-	-	-	
Species	*Acinetobacter_johnsonii*	3	0.06	1.83	*		-	-	-			-	-	-	
	*Acinetobacter*_species	-	-	-			1	8.57	1.6	*		-	-	-	
	*Pseudomonas_alcaliphila*	3	1.63	1.03	*		-	-	-			-	-	-	
	*Staphylococcus*_species	-	-	-			1, 3	1.6	7.24	*		-	-	-	
	*Vibrionimonas_magnilacihabitans*	3	1.63	0.23	*		-	-	-			-	-	-	
Genus	*Enterococcus*	1	0.37	5.34	*		-	-	-			-	-	-	
	*Jeotgalicoccus*	-	-	-			3	2.73	1.96	*		-	-	-	
	*Staphylococcus*	-	-	-			1, 3	2.29	9.98	*		-	-	-	
	*Vibrionimonas*	3	1.65	0.24	*		-	-	-			-	-	-	

^1^ Methods: method showing the differential abundance: 1, DESeq2; 2, MetagenomeSeq; 3, Metacoder. ^2^ Significance: *, *p* < 0.05; +, *p* < 0.1.

**Table 3 microorganisms-11-00673-t003:** Number, fraction of the diversity and abundance of ASVs shared between teat suspensions and milk samples from individual samples (C = 9; LY = 10) according to the group and the period.

	C ^1^			LY ^1^			*p*-Value ^3^	
	P1 ^2^	P2 ^2^	P3 ^2^	P1 ^2^	P2 ^2^	P3 ^2^	Group	Period
Fungi								
Number of cows	9	9	6	10	10	6		
Mean number of teat ASVs total	46	44	40	36	42	44	ns	ns
Mean number of milk ASVs total	13	16	9	12	18	11	ns	**+**
Mean number of ASVs shared by individual milk and teat	4	7	5	4	6	6	ns	**ns**
Mean fraction of milk ASVs shared with teat (%)	37	44	51	33	34	65	ns	******
Mean cumulative abundance in milk of shared ASVs (%)	35	68	65	36	28	47	ns	**+**
Bacteria								
Number of cows	9	9	9	10	9	9		
Mean number of teat ASVs total	113	107	70	104	103	74	ns	*******
Mean number of milk ASVs total	70	63	52	74	65	54	ns	******
Mean number of ASVs shared by individual milk and teat	47	34	25	47	33	27	ns	*******
Mean fraction of milk ASVs shared with teat (%)	67	54	44	64	50	50	ns	*******
Mean cumulative abundance in milk of shared ASVs (%)	89	54	57	82	71	74	ns	******

^1^ C = control group; LY = live yeast group. ^2^ P1 = before yeast administration; P2 = after three months of yeast supplementation; P3 = after four months of yeast supplementation. ^3^
*p*-Value: ***, *p* < 0.001; **, *p* < 0.01; +, *p* < 0.1; ns, non significant.

## Data Availability

Raw sequence data, metadata files and phyloseq objects are available at https://doi.org/10.57745/2FNJOG on 30 January 2023.

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
