# Peer review of "Dietary Live Yeast Supplementation Influence on Cow’s Milk, Teat and Bedding Microbiota in a Grass-Diet Dairy System"

_microorganisms, 2023, doi:10.3390/microorganisms11030673_

Round 1
Reviewer 1 Report
The manuscript illustrated the influence of supplemented live yeast on the performance and microbiota of raw milk, teat skin, and bedding material of dairy cows during different time, which of the results were interesting. However, the discussion part needs to be improved. Some information in materials and method was not clear. Here were the specific comments.
Materials and methods
Line 92 why do you choose yeast Saccharomyces cerevisiae CNCM I-1077 as the supplement?
Line 107 The individual milk (n=39), the teat skin (n=19) as well as the bedding material (n=2). Line 137 Statistical analysis was performed on 34 cows in total (17 per group, after removal of 2 outliers in C and 3 outliers in LY group).
But figure 1 (a) showed the control (n=9), LY(n=10) for milk sample. Figure 2 (a) showed each 2 bedding material samples for control and LY in P2, each 3 samples for control and LY in P3. Table 2 showed milk (n=20/group), teat (n=10/group), and bedding material (n=3/group). Why the sample numbers are different? Please explain the reason or clarify more clearly.
Result
Line 158-168 the significant difference between Control and LY was showed in Figure S1, which was a very important finding in this study. Therefore I recommend you put figure S1 not in supplement materials but in the manuscript text for better reading.
Line table 1 please write the full spelling for SEM. If it stands for Standard Error of Mean, I think there should be one SEM for parameters in Control and another one SEM for LY.
Line 177-178 A LY supplementation x week interaction was significant for all parameters (P<0.01). What data can support this conclusion?
Line 222 In figure 1 (a), the p-value for period was for P1 and P2, P1 and P3 or P1 and P2? Please clarify.
Line 259 figure 3(c) showed relative abundance of the fungal genera in the bedding material in P1 and P3, why not P2? Why only for fungi, not for bacteria?
Line 267 I did not see table caption for Methods1 and Sign.2, please add it.
Discussion
Line 335-337 Ruminal microbial populations were not studied in this experiment, but the expected outcomes in milk production were noted and confirmed the LY effect. Recommend add the Ruminal microbial analysis to explain your result in this part.
Line 394-397 The discussion of correlations between bacterial and fungal ASVs for two groups was not enough.
Conclusion
Line 406-407 Interesting changes in the microbial balance of three different components in the dairy cow environmental were revealed in particular by the individual analysis at the ASV level. The interesting changes should be more specific.
Author Response
Response to the comments of Reviewer 1
Comments and Suggestions for Authors
The manuscript illustrated the influence of supplemented live yeast on the performance and microbiota of raw milk, teat skin, and bedding material of dairy cows during different time, which of the results were interesting. However, the discussion part needs to be improved. Some information in materials and method was not clear. Here were the specific comments.
Answer: We thank the reviewer for the careful reading of the manuscript and for suggesting some revisions which help to improve our manuscript. The discussion has been completed. The materials and methods have been revised to be clearer. The changes in the manuscript are visible in revision mode and we refer to new line numbers below in our point-by-point response.
Materials and methods
Line 92 why do you choose yeast Saccharomyces cerevisiae CNCM I-1077 as the supplement?
Answer: We chose this strain of yeast because it is currently marketed for dairy cows as feed additive, with proven positive results on performance and rumen functions (see De Ondarza et al. 2010; Chaucheyras-Durand et al. 2016 in the reference list); however, its impact on milk microbiota is still poorly known in particular with the type of diet and production system like the ones we studied here.
Line 107 The individual milk (n=39), the teat skin (n=19) as well as the bedding material (n=2). Line 137 Statistical analysis was performed on 34 cows in total (17 per group, after removal of 2 outliers in C and 3 outliers in LY group).
But figure 1 (a) showed the control (n=9), LY(n=10) for milk sample. Figure 2 (a) showed each 2 bedding material samples for control and LY in P2, each 3 samples for control and LY in P3. Table 2 showed milk (n=20/group), teat (n=10/group), and bedding material (n=3/group). Why the sample numbers are different? Please explain the reason or clarify more clearly.
Answer:We agree with this remark. The Materials and Methods (Lines 110, 125 and 129) as well as the different tables and Figures have been modified to be more precise about the number of samples analyzed.
Result
Line 158-168 the significant difference between Control and LY was showed in Figure S1, which was a very important finding in this study. Therefore I recommend you put figure S1 not in supplement materials but in the manuscript text for better reading.
Answer: Thank you for the suggestion. The Figure 1 was added in Line 199.
Line table 1 please write the full spelling for SEM. If it stands for Standard Error of Mean, I think there should be one SEM for parameters in Control and another one SEM for LY.
Answer: SEM stand well for Standard Error of Mean as well as the example below. It corresponds to the overall variability of all samples in the statistical model
- Dairy Sci. 106:937–953 https://doi.org/10.3168/jds.2022-21851
Line 177-178 A LY supplementation x week interaction was significant for all parameters (P<0.01). What data can support this conclusion?
Answer: The p-value of the week by LY diet interaction being P<0.01 for all parameters, for the sake of clarity we thought it better not to add a column where all rows would have P<0.01, hence the asterisk and its explanation at the bottom of the table.
Line 222 In figure 1 (a), the p-value for period was for P1 and P2, P1 and P3 or P1 and P2? Please clarify.
Answer: Figure 1(a), now Figure 2 (a), shows the results of processing the microbial enumeration data by a multifactor Anova with group and period as factors. The p-value corresponds to the overall effect of the period and not a pairwise treatment.
Line 259 figure 3(c) showed relative abundance of the fungal genera in the bedding material in P1 and P3, why not P2? Why only for fungi, not for bacteria?
Answer: We presented the results of P1 and P3 because of the marked differences between the two bedding areas C and LY after 4 months of yeast supplementation in the animals' ration. In P2, the results were less marked between the two litters. Figure 3, now Figure 4, has been modified to show only the fungal data. Indeed, yeast supplementation mainly impacted the fungal populations, hence the choice of this presentation. Bacterial alpha diversity is now present in Table S3.
Line 267 I did not see table caption for Methods1 and Sign.2, please add it.
Answer: Indeed, the captions of Table 2 were missed. We added them.
Discussion
Line 335-337 Ruminal microbial populations were not studied in this experiment, but the expected outcomes in milk production were noted and confirmed the LY effect. Recommend add the Ruminal microbial analysis to explain your result in this part.
Answer: indeed, we did not look at rumen microbial populations in this study. As highlighted in new lines 365-366, the impact of the live yeast on milk production parameters can be linked to beneficial effects on rumen microbiota. Indeed, through a better ruminal microbial fermentation, energy supply to the cow is increased which leads to higher milk yield. The recent publications by Matthews et al. (2019) and Huang et al. (2021) nicely show connections between rumen microbiome and animal performance. A sentence has been added with the aforementioned references to complete our discussion (Lines 391-392).
Line 394-397 The discussion of correlations between bacterial and fungal ASVs for two groups was not enough.
Answer: The discussion has been completed in Lines 451-459.
Conclusion
Line 406-407 Interesting changes in the microbial balance of three different components in the dairy cow environmental were revealed in particular by the individual analysis at the ASV level. The interesting changes should be more specific.
Answer: The conclusion has been completed in Lines 470-471.

Reviewer 2 Report
The manuscript in hand describes effects of supplementation of live yeast to cows. Especially milk, bedding and teat skin were sampled for microbes, as well as parameters like milk production and other physical parameters auf the animals recorded. It seemed that the yeast supplemented was not found in milk, but rather changed the occurrence of other yeasts in the bedding material. Cows supplement with yeast had higher body weight and higher milk production.
Generally, it is a good idea to measure the impact of probiotic microorganisms (a term mostly avoided here) on the animals fed, i.e., cows. However, the control group received the carrier only , while the test group received yeast. 2 to 4 x 10^9 yeast cell are about 1 g, thus, the targeted amount of about 10^10 yeast cells per day are roughly 3.5 g. Does this amount makes a difference in providing additional nutrients, e.g., B vitamins? I do not know if this could be the reason for affecting body weight and milk production. Since I am not a cow expert, you might briefly mention this in your reply (or in the manuscript if appropriate).
Line 61: metabarcoding is a term used different in certain parts of biology. I would rather use “target gene sequencing” or "amplicon sequencing" instead.
Line 99: Who is Lallemand? Is this an important information? Is this a company? Then give city and country. (also line 103, Agrolab – here it is more clear because of name and circumstance).
Methods
While the method sections indeed lists all measurements and protocols used, it is – in my opinion – not detailed enough. At least, a method or protocol should be briefly mentioned (example: line 119-120: “bacteria were enumerated as described by Monsallier et al.” – While it is good to have a reference, enumeration could be plate counts, qPCR, FACS analyses or anything else. – this is just one example, there are more such instances before and after).
DNA-extraction: With our without bead beating? This has major influence.
Primer for 16S V-regions and ITS2 regions must be given. Primer cause major bias and, thus, they should be revealed in each publication.
Sequence on Illumina: I suspect v3 cartridges?
I know it is tedious to describe all the parameters in details, but they do matter (e.g., see here as an example: https://doi.org/10.1128/mSphere.01202-20).
Line 147 – 148: How (with which program?) the ASV sequences were matched to S. cerevisiae CNCM I-1077?
Here I also have the question on whether other yeast strains have the same or a different ITS2. I do not know the species-wide ITS2 difference for S. cerevisiae. This information helps judging on whether your finding really is “your” yeast or something else.
Bray-Curtis distance is “Bray-Curtis dissimilarity”. In any case, as far as I have understood, C and LY cows shared the same barn? In this case, I would suggest using generalized UniFrac, which is not just abundance of each ASV, but includes phylogenetic distances. The assumption is that a closely related strain in cow 1 will have a similar function in cow 2. Bray-Curtis dissimilarity assumes each ASV to be independent (i.e., unrelated), which is not wrong, but generalized UniFrac seems to be (much) more appropriate in your case.
DeSeq2 – why it is not a good idea, please see DOI: 10.1111/1755-0998.13730. Admittedly, I have not known about MetaGenomeSeq nor MetaCoder, but perhaps similar issues exist. I suggest using methods as described here DOI 10.7717/peerj.2836.
line 155: You filtered everything below 1%, correct? Congrats, few people filter and this is a good idea. 1% is even a bit strong, we use 0.25% but never mind.
Table 1
We got so used to our p-values if 0.05 and 0.01 that we do not think about them much further. Indeed, only lactose amount is significantly different, but low p-values are found for milk yield, milk fat, and casein. This, combined with body weight, makes a (weak, but observable) biological signal. (While you tell the reader in the abstract, there is not difference in body weight).
Line 177 (and in Table 1): I am not completely understood the notion “LY supplementation x week interaction” or “LY x week interaction”. Are you talking about a combined variable, or the multiplication of two variables, or LY over the time course of weeks? Sorry for being ignorant, but I think you should rephrase here.
Cell counts are given in log10. I know that some people / labs are doing this, while I believe the majority is not using log units. Why not 4x10^9 instead of 9.6 log10? And 5x10^9 instead of 9.7 log10? (lines 191ff and elsewhere). Well, if you like to stick to log-units, I will not insist.
Figure 1a – which actually is a table, and not a figure – Please round to the first position after the decimal point. There is no information in the second position after the decimal for count values, since your standard deviation is much larger than a fraction of 1/100.
Now in Fig. 2b, you give CFU/ml. (Of note, is it not an international understanding that CFU are always per ml (or g)?) – well, here you give CFU and not log-values, correct? Only use one notion, and I would suggest not using log-transformed CFU values.
Figure 2: use boxplots or violin plots. The current figure I find unusual. Or does single animals matter? I don’t think so.
Concerning Shannon index, I suggest using Shannon effective numbers, as explained here: "For the Shannon index, the relationship between species richness and the diversity is non-linear(!!!): at higher levels of species richness, communities will appear more similar (in terms of the magnitude of the index) that at lower levels of species richness. This is exactly how the index is supposed to behave, but ecologists are used to thinking about species richness, which behaves differently. Thus, Jost points out in a 2010 paper that it is so very tempting to misinterpret these indices, especially at high levels of richness. So what’ s the solution? Convert to effective numbers! (cited from https://jonlefcheck.net/2012/10/23/diversity-as-effective-numbers/)" – the three exclamation marks I have added. The non-linearity prevent also using standard statistical tests, which assume Poisson distributions and / or linear behavior!
Figure 3: Simpson appears here, but not in Methods. Same issue as before with Shannon. Use Simpson effective numbers, please.
Figure 3, panel b: The distance marker is missing in the plot (see Figure 1 in DOI 10.7717/peerj.2836). The p-value only asserts us, that at least one group shown is different from another, but not which pair causes the p-value. You should test pairwise, in order to find the significant different groups. Furthermore, P1, P2, and P3 are not independent, but connected since the same animals were measured 3-times and the microbiome of each animal should be similar between time points. Here, it becomes clear that it is even more advised to use generalized UniFrac, especially since the microbiome of the same animal should be similar in species composition, which should not be disregarded (as BC does).
Figure 3c: Using very similar colors in the color scale next to each other makes in extremely hard to see, which genus is which color. I had to use a graphics program to find out that the green block in control P2 and P3 is Galactomyces and not Mucor. I rather suggest contrasting colors for genera which touch each other in the column diagram. (of note, resolution of this panel is insufficient).
Table 2 – Not enough explanations. What is meant by “Methods1”? “Sign.2” is given elsewhere, but it should be given here again. Round to 1 position after the decimal. --- This table could be supplement and the distances between each group be shown as tree or NMDS plot for a better overview.
I am not sure what Fig. 4a should tell me. Similar numbers of ASVs between C and LY? Is this not expected? I would rather omit Fig. 4 a / b. Of note, write “fungi” and “bacteria” as headlines above the Venn diagrams.
Concerning panel (c), I assume the C for control was mistaken as panel C designation? At least “(c)” is missing.
Figure 4 c has insufficient resolution and I could not read the names. In any case, do you have a hypothesis for the correlations occurring? Otherwise it is just a nice colorful figure, but what is the “take-away message” of panel 4c?
Table 3. Does it really has information to the reader that the mean was 45.67? Could you not just round to 46? Especially since no fractions of ASVs exist and the past digit numbers are not telling us anything important.
Lines 307/308: Always add “S.” before species names, e.g., “S. hominis” instead of “hominis”.
Paragraph 3.8 => Genera and species names are not italic. Of note, as an exception of the rule, for bacteria and fungi also the family taxon is written italics, while for plants and animals it is not.
Discussion: you discuss Pichia as to be different between C and LY. However, in Fig. 3, colors in the color scale for Pichia do not match that in the columns. Pichia in the color scale has an RGB of 246/209/112, while the block I assume should be Pichia has 241/203/114. Not very much off, but since I had a hard time to find out, which one is which, this should be corrected with contrasting colors (see above).
Generally, please streamline your manuscript to match your aims more clearly. As far as I have understood, your aims are: 1) Is LY good for the cows? 2) Has LY an influence on the milk? (since it is used for PDO cheese making). Answer for 1) is Yes => more cow and milk, for 2) is No – which in your case is good. Did I overlook an aim? The last sentence in the abstract (“For the first time… “) seems to be a distraction (as, e.g., is Fig. 4). Or you need to spell out the third aim more clearly.
Unfortunately, I could not access raw data. I would be curious to run it with the tools I am used to. Concerning fungi, you might try https://sbi.hki-jena.de/daniel/latest/.
Author Response
Response to the comments of Reviewer 2
The manuscript in hand describes effects of supplementation of live yeast to cows. Especially milk, bedding and teat skin were sampled for microbes, as well as parameters like milk production and other physical parameters auf the animals recorded. It seemed that the yeast supplemented was not found in milk, but rather changed the occurrence of other yeasts in the bedding material. Cows supplement with yeast had higher body weight and higher milk production.
We thank the reviewer for the careful reading and for the suggestions for improving the manuscript. The modifications in the manuscript are available by revision mode and we refer to new line numbers below in our point-by-point response.
Generally, it is a good idea to measure the impact of probiotic microorganisms (a term mostly avoided here) on the animals fed, i.e., cows. However, the control group received the carrier only , while the test group received yeast. 2 to 4 x 10^9 yeast cell are about 1 g, thus, the targeted amount of about 10^10 yeast cells per day are roughly 3.5 g. Does this amount makes a difference in providing additional nutrients, e.g., B vitamins? I do not know if this could be the reason for affecting body weight and milk production. Since I am not a cow expert, you might briefly mention this in your reply (or in the manuscript if appropriate).
Answer: Thanks for this comment. The term probiotic as defined by the FAO/WHO and the ISAPP, which corresponds to “live microorganisms which when administered in adequate amounts confer a health benefit on the host” (Hill, C., Guarner, F., Reid, G. et al. The International Scientific Association for Probiotics and Prebiotics consensus statement on the scope and appropriate use of the term probiotic. Nat Rev Gastroenterol Hepatol 11, 506–514 (2014). https://doi.org/10.1038/nrgastro.2014.66) is appropriate in the case of the live yeast administred here, as this strain I-1077 has been reported to exhibit positive effects on the host animal by promoting rumen microbiota activity. The commercial product is very concentrated and accounts for 10 x 109 CFU per gram. In this study, the product is distributed though a pelleted concentrate to bring daily 1g/cow, i.e., 10 x 109 CFU. Through this daily supplementation, rumen fermentation is more efficient and better feed conversion occurs, leading to an improved performance of dairy cows (milk production in particular). This is supported by a range of peer reviewed publications, some of them being listed in the manuscript (De Ondarza et al., 2010; Bach et al., 2008; Chaucheyras-Durand et al., 2016). The suggested modes of action are rather interactions between live yeast cells and the resident rumen microbiota, with a positive orientation of microbial balance, and not the direct supply of nutrients to the animal. The impact of this additive on rumen microbial fermentation was mentioned at the beginning of introduction and at the start of discussion.
Line 61: metabarcoding is a term used different in certain parts of biology. I would rather use “target gene sequencing” or "amplicon sequencing" instead.
Answer: In the reference article of Berg et al. (Berg, G., Rybakova, D., Fischer, D. et al. Microbiome definition re-visited: old concepts and new challenges. Microbiome 8, 103 (2020). https://doi.org/10.1186/s40168-020-00875-0) the term « metabarcoding » is addressed as followed: “Currently available methods for studying microbiomes, so-called multi-omics, range from high throughput isolation (culturomics) and visualization (microscopy), to targeting the taxonomic composition (metabarcoding), or addressing the metabolic potential (metabarcoding of functional genes, metagenomics) to analyze microbial activity (metatranscriptomics, metaproteomics, metabolomics)”. Please see also the figure below. We feel that the term metabarcoding we used does clearly correspond to what we have reported, so we would like to keep this term in the manuscript.
Line 99: Who is Lallemand? Is this an important information? Is this a company? Then give city and country. (also line 103, Agrolab – here it is more clear because of name and circumstance).
Answer: Lallemand is a company that has been involved in this study by providing the live yeast additive. We added the missing information on city and country in Lines 96 and 106.
Methods
While the method sections indeed lists all measurements and protocols used, it is – in my opinion – not detailed enough. At least, a method or protocol should be briefly mentioned (example: line 119-120: “bacteria were enumerated as described by Monsallier et al.” – While it is good to have a reference, enumeration could be plate counts, qPCR, FACS analyses or anything else. – this is just one example, there are more such instances before and after).
Answer: The enumeration of the microbial groups were on plate counts with different culture media. It has been added in Line 123.
DNA-extraction: With our without bead beating? This has major influence.
Answer: As preconized by the FastDNA Spin Kit for Soil we use a Precellys Evolution homogenizer. We added the information in Lines 135.
Primer for 16S V-regions and ITS2 regions must be given. Primer cause major bias and, thus, they should be revealed in each publication.
Answer: It has been done in Lines 137-149.
Sequence on Illumina: I suspect v3 cartridges?
Answer: It has been done in L150.
I know it is tedious to describe all the parameters in details, but they do matter (e.g., see here as an example: https://doi.org/10.1128/mSphere.01202-20).
Answer: The Material and Methods section has been expanded to provide more details.
Line 147 – 148: How (with which program?) the ASV sequences were matched to S. cerevisiae CNCM I-1077?
Answer: Sequences were compared using blastn with 100% identity and 100% coverage. Details have been provided in the Materials and Methods and Results sections on lines 160 and 230 respectively.
Here I also have the question on whether other yeast strains have the same or a different ITS2. I do not know the species-wide ITS2 difference for S. cerevisiae. This information helps judging on whether your finding really is “your” yeast or something else.
Answer: We agree that the ITS based amplicon sequencing approach probably does not allow to monitor S. cerevisiae at the strain level. However as we reported in the article in Lines 227-229.
Bray-Curtis distance is “Bray-Curtis dissimilarity”. In any case, as far as I have understood, C and LY cows shared the same barn? In this case, I would suggest using generalized UniFrac, which is not just abundance of each ASV, but includes phylogenetic distances. The assumption is that a closely related strain in cow 1 will have a similar function in cow 2. Bray-Curtis dissimilarity assumes each ASV to be independent (i.e., unrelated), which is not wrong, but generalized UniFrac seems to be (much) more appropriate in your case.
Answer: For the betadiversity analysis, two methods were used, Bray-Curtis dissimilarity and generalized Unifrac whose p-values are presented in Table S4. The conclusions are the same regardless of the method. Furthermore, we compared open environments (milk, teat and bedding material) for which it is not clear that the individual effect is dominant.
DeSeq2 – why it is not a good idea, please see DOI: 10.1111/1755-0998.13730. Admittedly, I have not known about MetaGenomeSeq nor MetaCoder, but perhaps similar issues exist. I suggest using methods as described here DOI 10.7717/peerj.2836.
Answer: Although DeSeq2, metagenomeSeq and metacoder are widely used in the field of amplicon data for differential analysis, they all have their limitations and prerequisites. Each method use a different type of normalisation and their own way to handle the sparsity of those data. As mention in doi:10.1111/1755-0998.13730 DeSeq2 can over estimate p-values due to the highly dispersion of the data."MetaGenomeSeq has been designed with microbial marker surveys in mind, using a normalization procedure and a zero-inflated gaussian (ZIG) mixture model." (cited from doi:10.1186/s40168-016-0208-8). MetaCoder use a simple and classic non-parametric Kruskal-Wallis test which is the same test used in the package you mentionned (Rhea). We do agree sparsity of our tables is the main problem for our type of data, and this is why we use three different methods to correlate results as well as a strong 1% filtering as you mention below
line 155: You filtered everything below 1%, correct? Congrats, few people filter and this is a good idea. 1% is even a bit strong, we use 0.25% but never mind.
Answer: see above
Table 1
We got so used to our p-values if 0.05 and 0.01 that we do not think about them much further. Indeed, only lactose amount is significantly different, but low p-values are found for milk yield, milk fat, and casein. This, combined with body weight, makes a (weak, but observable) biological signal. (While you tell the reader in the abstract, there is not difference in body weight).
Answer: Thank you for highlighting this point. The abstract has been modified.
Line 177 (and in Table 1): I am not completely understood the notion “LY supplementation x week interaction” or “LY x week interaction”. Are you talking about a combined variable, or the multiplication of two variables, or LY over the time course of weeks? Sorry for being ignorant, but I think you should rephrase here.
Answer: "LY x week interaction" is not a combined variable. We tested the interaction between two factors (week and supplementation) to answer the question "is the effect of yeast variable over time?"
Cell counts are given in log10. I know that some people / labs are doing this, while I believe the majority is not using log units. Why not 4x10^9 instead of 9.6 log10? And 5x10^9 instead of 9.7 log10? (lines 191ff and elsewhere). Well, if you like to stick to log-units, I will not insist.
Answer: the log10 transformation is a commonly used way of expressing microbiological counts so we have assumed that the readers of Microorganisms are well trained to understand this unit.
Figure 1a – which actually is a table, and not a figure – Please round to the first position after the decimal point. There is no information in the second position after the decimal for count values, since your standard deviation is much larger than a fraction of 1/100.
Answer: The round to 1 decimal has been done as suggested in the new Figure 2
Now in Fig. 2b, you give CFU/ml. (Of note, is it not an international understanding that CFU are always per ml (or g)?) – well, here you give CFU and not log-values, correct? Only use one notion, and I would suggest not using log-transformed CFU values.
Answer: We have changed the CFU/mL to log10 CFU/mL in the new Figure 1(b) to be consistent with the table in Figure 2(a).
Indeed, these are CFU/mL of milk. If we had counted the populations in cheese for example, the unit of count would have been CFU/g.
Figure 2: use boxplots or violin plots. The current figure I find unusual. Or does single animals matter? I don’t think so.
Answer: We preferred this type of presentation to boxplots to emphasise the differences between individuals.
Concerning Shannon index, I suggest using Shannon effective numbers, as explained here: "For the Shannon index, the relationship between species richness and the diversity is non-linear(!!!): at higher levels of species richness, communities will appear more similar (in terms of the magnitude of the index) that at lower levels of species richness. This is exactly how the index is supposed to behave, but ecologists are used to thinking about species richness, which behaves differently. Thus, Jost points out in a 2010 paper that it is so very tempting to misinterpret these indices, especially at high levels of richness. So what’ s the solution? Convert to effective numbers! (cited from https://jonlefcheck.net/2012/10/23/diversity-as-effective-numbers/)" – the three exclamation marks I have added. The non-linearity prevent also using standard statistical tests, which assume Poisson distributions and / or linear behavior!
Answer: The individual alphadiversity data are made available as supplementary data in Table S5 to allow anyone to make further calculations.
Figure 3: Simpson appears here, but not in Methods. Same issue as before with Shannon. Use Simpson effective numbers, please.
Answer: There was a mistake: we only speak about Shannon
Figure 3, panel b: The distance marker is missing in the plot (see Figure 1 in DOI 10.7717/peerj.2836). The p-value only asserts us, that at least one group shown is different from another, but not which pair causes the p-value. You should test pairwise, in order to find the significant different groups. Furthermore, P1, P2, and P3 are not independent, but connected since the same animals were measured 3-times and the microbiome of each animal should be similar between time points. Here, it becomes clear that it is even more advised to use generalized UniFrac, especially since the microbiome of the same animal should be similar in species composition, which should not be disregarded (as BC does).
Answer: As indicated in the legend of Figure 3, now Figure 4, a pairwise test was performed: all combinations had a p-value<0.05
Figure 3c: Using very similar colors in the color scale next to each other makes in extremely hard to see, which genus is which color. I had to use a graphics program to find out that the green block in control P2 and P3 is Galactomyces and not Mucor. I rather suggest contrasting colors for genera which touch each other in the column diagram. (of note, resolution of this panel is insufficient).
Answer: we have modified the graph in the new Figure 4(c) to make it more readable
Table 2 – Not enough explanations. What is meant by “Methods1”? “Sign.2” is given elsewhere, but it should be given here again. Round to 1 position after the decimal. --- This table could be supplement and the distances between each group be shown as tree or NMDS plot for a better overview.
Answer: Indeed, the captions of Table 2 were missed. We added them. Rounding to 1 decimal would result in a value of 0 for some taxa present in very low abundance. It seems important to us to make the difference between 0 and 0.03 for example. We therefore prefer not to round to 1 decimal.
In our opinion, it is preferable to keep Table 2 as it is, as it allows the three compartments to be compared simultaneously and already contains only those taxa whose abundance exceeds 1% and which are differentially abundant between C and LY. For ease of reading, we propose to replace the empty boxes with "-" (dash 6).
I am not sure what Fig. 4a should tell me. Similar numbers of ASVs between C and LY? Is this not expected? I would rather omit Fig. 4 a / b. Of note, write “fungi” and “bacteria” as headlines above the Venn diagrams.
Answer: Figure 4, now Figure 5, shows the fungal (a) and bacterial (b) ASVs shared between the different compartments under the C or LY regime. It can be seen that the number of shared ASVs was not influenced by the diet, which was the objective of the experiment. We prefer to keep this figure. For a better reading we added "fungi" and "bacteria" as titles above the Venn diagrams as suggested.
Concerning panel (c), I assume the C for control was mistaken as panel C designation? At least “(c)” is missing.
Figure 4 c has insufficient resolution and I could not read the names. In any case, do you have a hypothesis for the correlations occurring? Otherwise it is just a nice colorful figure, but what is the “take-away message” of panel 4c?
Answer: As proposed, the Figure 4(c) has been removed.
Table 3. Does it really has information to the reader that the mean was 45.67? Could you not just round to 46? Especially since no fractions of ASVs exist and the past digit numbers are not telling us anything important.
Answer: As suggested the numbers were rounded.
Lines 307/308: Always add “S.” before species names, e.g., “S. hominis” instead of “hominis”.
Answer: We modified the text as suggested in L
Paragraph 3.8 => Genera and species names are not italic. Of note, as an exception of the rule, for bacteria and fungi also the family taxon is written italics, while for plants and animals it is not.
Answer: Indeed, we agree. It was an oversight that has been corrected.
Discussion: you discuss Pichia as to be different between C and LY. However, in Fig. 3, colors in the color scale for Pichia do not match that in the columns. Pichia in the color scale has an RGB of 246/209/112, while the block I assume should be Pichia has 241/203/114. Not very much off, but since I had a hard time to find out, which one is which, this should be corrected with contrasting colors (see above).
Answer: we have modified the new Figure 4 (c) to make it more readable
Generally, please streamline your manuscript to match your aims more clearly. As far as I have understood, your aims are: 1) Is LY good for the cows? 2) Has LY an influence on the milk? (since it is used for PDO cheese making). Answer for 1) is Yes => more cow and milk, for 2) is No – which in your case is good. Did I overlook an aim? The last sentence in the abstract (“For the first time… “) seems to be a distraction (as, e.g., is Fig. 4). Or you need to spell out the third aim more clearly.
Answer: This study showed that 1) the yeast supplementation in the cows' ration had no impact on the milk microbiota and 2) the introduced strain was not transferred to the milk since its sequence was not identified in the milk, which is important for use in the PDO cheese industry. In addition, the supplementation in S. cerevisiae has a beneficial effect on milk production parameters. The abstract and the Figure 4, now Figure 5, were modified.
Unfortunately, I could not access raw data. I would be curious to run it with the tools I am used to. Concerning fungi, you might try https://sbi.hki-jena.de/daniel/latest/.
Reviewer 3 Report
Th author study Saccharomyces cerevisiae supplementation in dairy cow on milk yield, fungal and bacteria population in milk, teat, and bedding materials.
Line 17: Please explain the connection of microbial additives on raw milk cheese quality to raw milk microbiota.
Line 21 Which stage of dairy cattle was used for the trial?
Please specify the concentration of Yeast used in the trial.
Line 71 . Is energy level of grass matter in this study?
Line 107: Please explain why only 19 teat skin were collected at each time point.
Please specify the location of pen where bedding material were sampled. How far away is the sampling area from feed trough?
Line 130: Please specify how raw sequence were processed before statistical analysis
Line 160: Since an interaction of day and treatment were found on FCR, this statement here should be removed.
Line 165: Was milk yield higher in LY group at the beginning of trial? The text in material and methods suggest cows were stratified by milk yield during adaptation period?
Line 167: Please explain how the ECR was produced. In addition, it appears that variation of timepoints existed in traits measured, hence I would suggest to present the traits that approach significant day by treatment interaction in text while leave those insignificant traits as supplemental materials.
Line 181 Note that There was a time by treatment interaction in ripening bacteria that indicate the level of differences between two treatments were inconsistent among the times. Therefore, it is of important to provide the results to readers. Moreover, please explain why half of samples didn’t show the increase on Yeast after exposed to supplement.
Figure 2 Please double check if there was no mistake on relative abundance of Saccharomyces cerevisiae _-1077 in bedding materials and on teat. Results looks suspicious. For instance, 4601 teat had high abundant while remain samples in control group are almost nondetectable. Do we know why variation on abundant differs this much among treated group? Moreover, If number refers to each cow, why cow that showed higher count of Yeast in Figure 2b had lower abundance of Saccharomyces cerevisiae on teat or vice versa (2635, 2158, 2168 and etc)?
Line 208: Please also include P 1 data.
Line 214: Please provide R and P value of the comparison.
Table 2 Please define the asterisk and cross superscript in legend.
The table contain excess amount of information and is hard to follow. I would recommend present those results that are intriguing and corelative to objectives of studies and leave remain in supplemental materials.
Line 284. Does this indicate the milk samples devoid Saccharomyces cerevisiae and Saccharomyces cerevisiae in bedding material and teat were from diets?
Line 352 According to figure 2, results indicated that Saccharomyces cerevisiae can be spotted in teat of LY group, but variation between samples is the concern?
Author Response
Response to the comments of Reviewer 3
Comments and Suggestions for Authors
The author study Saccharomyces cerevisiae supplementation in dairy cow on milk yield, fungal and bacteria population in milk, teat, and bedding materials.
We thank the reviewer for the careful reading of the manuscript. The changes in the manuscript are visible in revision mode and we refer to new line numbers below in our point-by-point response
Line 17: Please explain the connection of microbial additives on raw milk cheese quality to raw milk microbiota.
Answer: It is important for traditional or quality-labelled raw milk cheeses that the animal feed, especially if it is supplemented with additives, does not have a negative impact on the final sensory and health quality of the product. Hence the purpose of our study.
Line 21 Which stage of dairy cattle was used for the trial? Please specify the concentration of Yeast used in the trial.
Answer: The stages of dairy cattle and the yeast concentration were added in the abstract.
Line 71 . Is energy level of grass matter in this study?
Answer: the low energy grass diet is characteristic from a semi-mountain production system we wanted to focus on in this study. The sentence has been modified to be more precise (Lines 73-74).
Line 107: Please explain why only 19 teat skin were collected at each time point.Please specify the location of pen where bedding material were sampled. How far away is the sampling area from feed trough?
Answer: The Materials and Methods has been modified to be more precise about the number of samples analyzed (Lines 110, 125 and 129) and how the bedding materials were sampled (Line 117).
Line 130: Please specify how raw sequence were processed before statistical analysis
Answer: It has been done in Lines 154-158
Line 160: Since an interaction of day and treatment were found on FCR, this statement here should be removed.
Answer: Indeed, it has been removed
Line 165: Was milk yield higher in LY group at the beginning of trial? The text in material and methods suggest cows were stratified by milk yield during adaptation period?
Answer: our sentence was maybe a little bit confusing. Milk yields were similar in both groups at the start of the trial thanks to balanced groups constituted at the end of adaptation period. We measured an increase in milk yeast rapidly after the start of LY supplementation. To improve clarity, we modified the sentence.
Line 167: Please explain how the ECR was produced. In addition, it appears that variation of timepoints existed in traits measured, hence I would suggest to present the traits that approach significant day by treatment interaction in text while leave those insignificant traits as supplemental materials.
Answer: there is a typo in the table, it should be read ECM for energy corrected milk. This ECM is calculated with the following equation:
ECM (kg/d) = 0.3246 * milk yield (kg) + 12.86 * %fat + 7.04 * %protein
According to Tyrrell and Reid, 1965 (doi.org/10.3168/jds.S0022-0302(65)88430-2)
To our opinion, it is preferable to keep all measures in the main manuscript (Table 1), because they reflect the important parameters that are needed to characterize animal performance.
Line 181 Note that There was a time by treatment interaction in ripening bacteria that indicate the level of differences between two treatments were inconsistent among the times. Therefore, it is of important to provide the results to readers. Moreover, please explain why half of samples didn’t show the increase on Yeast after exposed to supplement.
Answer: Indeed, a period by group interaction in the ripening bacteria was statistically observed. However, since the variations do not exceed 0.5 log, it is considered in cultural microbiology that this corresponds to biological variations. Yeasts of all genera and species were enumerated in individual animal milks by culture on OGA medium. Usually this enumeration is carried out on the mixed milks of several cows. In this trial, we have shown the fact that cows exposed to live yeast supplement do not all show an increase in total yeast counts are due to natural individual variability.
Figure 2 Please double check if there was no mistake on relative abundance of Saccharomyces cerevisiae _-1077 in bedding materials and on teat. Results looks suspicious. For instance, 4601 teat had high abundant while remain samples in control group are almost nondetectable. Do we know why variation on abundant differs this much among treated group? Moreover, If number refers to each cow, why cow that showed higher count of Yeast in Figure 2b had lower abundance of Saccharomyces cerevisiae on teat or vice versa (2635, 2158, 2168 and etc)?
Answer: The relative abundances of Saccharomyces cerevisiae CN-1077 in the bedding material and on the teats are correct. Indeed, the results may seem surprising but Figure 2, now Figure 3, shows the abundances of the ASV corresponding to the supplemented yeast, which were very low but varied from one individual to another. However, it can be seen that this ASV is more frequently found on LY teats, which seems consistent. The number does refer to the cow. However, the count results in Figure 1, now Figure 2, cannot be compared with those in Figure 2, now Figure3. For example, the E2158 cow had high levels of yeast by cultural method but it was probably not the introduced yeast. Finding the sequence does not imply a high level in culture.
Line 208: Please also include P 1 data.
Answer: It has been done in Figure 4(a) and in Table S3.
Line 214: Please provide R and P value of the comparison.
Answer: The Figure 3(c), now Figure 4(c), is a graphical representation of the abundance of fungal genera. It is not a statistical comparison.
Table 2 Please define the asterisk and cross superscript in legend.
Answer: It has been done (Lines 312-313)
The table contain excess amount of information and is hard to follow. I would recommend present those results that are intriguing and corelative to objectives of studies and leave remain in supplemental materials.
Answer: In our opinion, it is preferable to keep Table 2 as it is, as it allows the three compartments to be compared simultaneously and already contains only those taxa whose abundance exceeds 1% and which are differentially abundant between C and LY. For ease of reading, we propose to replace the empty boxes with "-" (dash 6).
Line 284. Does this indicate the milk samples devoid Saccharomyces cerevisiae and Saccharomyces cerevisiae in bedding material and teat were from diets?
Answer: Indeed. The objective of our study was to evaluate if the live yeast Saccharomyces cerevisiae supplemented diet to dairy cows influenced the microbiota of raw milk, teat skin and bedding material. The metabarcoding based on amplicon sequence variants (ASV) level for microbial community characterization provides a high resolution consistent with intrinsic biological meaning and so allowed us to track the sequence of S. cerevisiae CNCM I-1077 in our different compartments.
Line 352 According to figure 2, results indicated that Saccharomyces cerevisiae can be spotted in teat of LY group, but variation between samples is the concern?
Answer: Indeed, an inter-individual variation in the presence of the sequence corresponding to S. cerevisiae was observed. But its abundance remains low. In our study, we worked at the individual level, which is innovative to our knowledge, with the aim of evaluating the biological variation between individuals.
Reviewer 4 Report
Quality work, clear justification of hypotheses, rich literary research. The correct technological and laboratory methods were used, a sufficient number of individuals for experiments used. The results are well interpreted, reaching standard values. Statistical evaluation (suitable methods have been used) is processed at a high level, suitably complemented by tables and graphs. Everything is documented by a suitable discussion. The references correspond to the methods of the editors.
Author Response
Response to the comments of Reviewer 4
Comments and Suggestions for Authors
Quality work, clear justification of hypotheses, rich literary research. The correct technological and laboratory methods were used, a sufficient number of individuals for experiments used. The results are well interpreted, reaching standard values. Statistical evaluation (suitable methods have been used) is processed at a high level, suitably complemented by tables and graphs. Everything is documented by a suitable discussion. The references correspond to the methods of the editors.
We thank the reviewer for these positive comments.
Round 2
Reviewer 3 Report
Line 17: Please explain the connection of microbial additives on raw milk cheese quality to raw milk microbiota.
Answer: It is important for traditional or quality-labelled raw milk cheeses that the animal feed, especially if it is supplemented with additives, does not have a negative impact on the final sensory and health quality of the product. Hence the purpose of our study.
R: I am not disagreed with the importance of raw milk cheese labeling but do have concern on how this sentence relates to hypothesis of this study, especially when cheese quality was never being evaluated in this article?
Line 21 Which stage of dairy cattle was used for the trial? Please specify the concentration of Yeast used in the trial.
Answer: The stages of dairy cattle and the yeast concentration were added in the abstract.
R: Please point out where these information are from Line number. I can’t find this information?
Line 24-25: Does this mean BW increase is inversely correlated with milk yield. Therefore author use “but” to connect two sentences? In addition, BW of LY group was tended to be higher at the beginning of trial. That means the difference of BW at end of trial is not outcome of Yeast supplementation but carry over effect of starting BW. Thus the statement has to be rephrase.
Line 167: Please explain how the ECR was produced. In addition, it appears that variation of timepoints existed in traits measured, hence I would suggest to present the traits that approach significant day by treatment interaction in text while leave those insignificant traits as supplemental materials.
Answer: there is a typo in the table, it should be read ECM for energy corrected milk. This ECM is calculated with the following equation:
ECM (kg/d) = 0.3246 * milk yield (kg) + 12.86 * %fat + 7.04 * %protein
According to Tyrrell and Reid, 1965 (doi.org/10.3168/jds.S0022-0302(65)88430-2)
R: Please improve figure quality. It is blurry.
Line 201 How about the higher ECM during week 10 to 13?
Line 246 P as for P value should be italic, and space should be insert on both side of equation. This applies to whole manuscript.
Line 181 Note that There was a time by treatment interaction in ripening bacteria that indicate the level of differences between two treatments were inconsistent among the times. Therefore, it is of important to provide the results to readers. Moreover, please explain why half of samples didn’t show the increase on Yeast after exposed to supplement.
Answer: Indeed, a period by group interaction in the ripening bacteria was statistically observed. However, since the variations do not exceed 0.5 log, it is considered in cultural microbiology that this corresponds to biological variations. Yeasts of all genera and species were enumerated in individual animal milks by culture on OGA medium. Usually this enumeration is carried out on the mixed milks of several cows. In this trial, we have shown the fact that cows exposed to live yeast supplement do not all show an increase in total yeast counts are due to natural individual variability.
R: I do not aware there is considered a okay as long as variation remained below 0.5 log. In addition, some treatment groups had SE above 0.5. There is reason why author had data analyzed statistically at first place, and results should be described according to it. Otherwise what is reason to conduct statistic analysis anyway? These variation might be important to explain the observation on culture dependent and 16S amplicon results?
Also Please avoid overstatement of those traits that are not significant (line 251-253).
Line 601: How is Fiber able to colonized since it is not an organism?
Figure 2 Please double check if there was no mistake on relative abundance of Saccharomyces cerevisiae _-1077 in bedding materials and on teat. Results looks suspicious. For instance, 4601 teat had high abundant while remain samples in control group are almost nondetectable. Do we know why variation on abundant differs this much among treated group? Moreover, If number refers to each cow, why cow that showed higher count of Yeast in Figure 2b had lower abundance of Saccharomyces cerevisiae on teat or vice versa (2635, 2158, 2168 and etc)?
Answer: The relative abundances of Saccharomyces cerevisiae CN-1077 in the bedding material and on the teats are correct. Indeed, the results may seem surprising but Figure 2, now Figure 3, shows the abundances of the ASV corresponding to the supplemented yeast, which were very low but varied from one individual to another. However, it can be seen that this ASV is more frequently found on LY teats, which seems consistent. The number does refer to the cow. However, the count results in Figure 1, now Figure 2, cannot be compared with those in Figure 2, now Figure3. For example, the E2158 cow had high levels of yeast by cultural method but it was probably not the introduced yeast. Finding the sequence does not imply a high level in culture.
R: Samples from Teat and bedding material represented the environment rather the ingestion of yeast (line 515). It does appear that live yeast population increased in some milk samples from those that received the yeast treatment despite the variation. Was live Saccharomyces cerevisiae CN-1077 found in milk samples?
Why is the variation observed between P2 and P3 from teat suspension of LY group? I can relate to the purpose of showing individual result, but level of variation makes it hard to not notice. It also rise the question when correlate the yeast content in milk to phenotype responses.
Line 214: Please provide R and P value of the comparison.
Answer: The Figure 3(c), now Figure 4(c), is a graphical representation of the abundance of fungal genera. It is not a statistical comparison.
R: Pease check the SEM in figure 4a.
Line 281: Results indicated the teat exert quite different community structure across the time and group, and the same effect was not observed in milk samples. Again there were several fungi shared between teat and bedding material, but not found in milk? While two bacteria shared between teat and milk but not in bedding material. Is there any reason why shared bacteria can be found between teat and milk but not in fungi?
Line 582: Please remove positively in LY because they were not significant as indicated by author.
Author Response
Response to the comments of Reviewer 3 (Round 2)
Line 17: Please explain the connection of microbial additives on raw milk cheese quality to raw milk microbiota.
Answer: It is important for traditional or quality-labelled raw milk cheeses that the animal feed, especially if it is supplemented with additives, does not have a negative impact on the final sensory and health quality of the product. Hence the purpose of our study.
R: I am not disagreed with the importance of raw milk cheese labeling but do have concern on how this sentence relates to hypothesis of this study, especially when cheese quality was never being evaluated in this article?
Response R2: Indeed cheese quality was not evaluated in this study but milk microbial compositions were analyzed and served as a proxy to claim that milk was not impacted by dietary live yeast supplementation of the cows. The sentence on Line 16 has been modified like this : “The supplementation of animal feed with microbial additives remains questioning for the traditional or quality label raw milk cheese with regard to microbial transfer to milk”.
Line 21 Which stage of dairy cattle was used for the trial? Please specify the concentration of Yeast used in the trial.
Answer: The stages of dairy cattle and the yeast concentration were added in the abstract.
R: Please point out where these information are from Line number. I can’t find this information?
Response R2: Indeed we forgot to add the DIM in the abstract, it was corrected in Lines 20-21 and in Line 83. Yeast concentration is indicated in Lines 21 and 96.
Line 24-25: Does this mean BW increase is inversely correlated with milk yield. Therefore author use “but” to connect two sentences? In addition, BW of LY group was tended to be higher at the beginning of trial. That means the difference of BW at end of trial is not outcome of Yeast supplementation but carry over effect of starting BW. Thus the statement has to be rephrase.
Response R2: Indeed, the sentence formulation was not correct and “but” was replaced by “and”. Over the experiment, the BW increase was of 0.9kg only in control group and of 19.6kg in LY group for a difference of 12.7kg between the 2 groups at trial beginning. As the difference at start was not significant, we didn’t analyze the evolution of BW per se but just the initial and final points.
Line 167: Please explain how the ECR was produced. In addition, it appears that variation of timepoints existed in traits measured, hence I would suggest to present the traits that approach significant day by treatment interaction in text while leave those insignificant traits as supplemental materials.
Answer: there is a typo in the table, it should be read ECM for energy corrected milk. This ECM is calculated with the following equation:
ECM (kg/d) = 0.3246 * milk yield (kg) + 12.86 * %fat + 7.04 * %protein
According to Tyrrell and Reid, 1965 (doi.org/10.3168/jds.S0022-0302(65)88430-2)
R: Please improve figure quality. It is blurry.
Response R2: As suggested we improved the quality of the Figure 1.
Line 201 How about the higher ECM during week 10 to 13?
Response R2: During those 4 weeks, the evolution of ECM was quite variable between the two groups (trend to be lower on week 10, no difference on week 11, and significantly lower on weeks 12 and 13 in LY group compared to control, while milk yield was similar between both groups), whereas from week 14 until the end of experimental period, ECM was significantly and consistently higher in LY group compared to control, as was also milk yield, this is why we highlighted this result.
Line 246 P as for P value should be italic, and space should be insert on both side of equation. This applies to whole manuscript.
Response R2: Thank you for this remark. The modification has been done in the whole manuscript.
Line 181 Note that There was a time by treatment interaction in ripening bacteria that indicate the level of differences between two treatments were inconsistent among the times. Therefore, it is of important to provide the results to readers. Moreover, please explain why half of samples didn’t show the increase on Yeast after exposed to supplement.
Answer: Indeed, a period by group interaction in the ripening bacteria was statistically observed. However, since the variations do not exceed 0.5 log, it is considered in cultural microbiology that this corresponds to biological variations. Yeasts of all genera and species were enumerated in individual animal milks by culture on OGA medium. Usually this enumeration is carried out on the mixed milks of several cows. In this trial, we have shown the fact that cows exposed to live yeast supplement do not all show an increase in total yeast counts are due to natural individual variability.
R: I do not aware there is considered a okay as long as variation remained below 0.5 log. In addition, some treatment groups had SE above 0.5. There is reason why author had data analyzed statistically at first place, and results should be described according to it. Otherwise what is reason to conduct statistic analysis anyway? These variation might be important to explain the observation on culture dependent and 16S amplicon results?
Response R2: Indeed an interaction was observed and was mentioned in Line 208.
The total yeast were detected at 1.59 CFU/ml in Control group in P1 (2 from 9 cows), 2.5 cfu/ml and 4 cfu/ml in LY group in P2 (5 from 10 cows) and P3 (3 from 10 cows) respectively as shwon in Figure 1b. Those values are indeed numerically higher in LY than in Control but 4 yeasts/ml remain extremely low in terms of potential biological impact on milk quality, safety and cheese properties.
Also Please avoid overstatement of those traits that are not significant (line 251-253).
Response R2: As suggested, we deleted these not significant traits in Lines 281-283.
Line 601: How is Fiber able to colonized since it is not an organism?
Response R2: We agree that the sentence in Line 369 was not clear. It has been modified.
Figure 2 Please double check if there was no mistake on relative abundance of Saccharomyces cerevisiae _-1077 in bedding materials and on teat. Results looks suspicious. For instance, 4601 teat had high abundant while remain samples in control group are almost nondetectable. Do we know why variation on abundant differs this much among treated group? Moreover, If number refers to each cow, why cow that showed higher count of Yeast in Figure 2b had lower abundance of Saccharomyces cerevisiae on teat or vice versa (2635, 2158, 2168 and etc)?
Answer: The relative abundances of Saccharomyces cerevisiae CN-1077 in the bedding material and on the teats are correct. Indeed, the results may seem surprising but Figure 2, now Figure 3, shows the abundances of the ASV corresponding to the supplemented yeast, which were very low but varied from one individual to another. However, it can be seen that this ASV is more frequently found on LY teats, which seems consistent. The number does refer to the cow. However, the count results in Figure 1, now Figure 2, cannot be compared with those in Figure 2, now Figure3. For example, the E2158 cow had high levels of yeast by cultural method but it was probably not the introduced yeast. Finding the sequence does not imply a high level in culture.
R: Samples from Teat and bedding material represented the environment rather the ingestion of yeast (line 515). It does appear that live yeast population increased in some milk samples from those that received the yeast treatment despite the variation. Was live Saccharomyces cerevisiae CN-1077 found in milk samples?
Response R2 : We agree that total live yeast populations increase in some milk samples. Using metabarcoding approach we showed that this was associated with the increased relative abundance of the yeast Aspergillus in milk while no sequence of S. cerevisiae CNCM I-1077 was observed in milk. It suggests that this specific strain was not present in milks from LY cows.
Why is the variation observed between P2 and P3 from teat suspension of LY group? I can relate to the purpose of showing individual result, but level of variation makes it hard to not notice. It also rise the question when correlate the yeast content in milk to phenotype responses.
Response R2: The sequence with 100% similarity with that of S. cerevisiae CNCM I-1077 strain was detected at very low relative abundances, which may explain why it is detected sporadically depending on the individuals and the periods on the teats.
Line 214: Please provide R and P value of the comparison.
Answer: The Figure 3(c), now Figure 4(c), is a graphical representation of the abundance of fungal genera. It is not a statistical comparison.
R: Pease check the SEM in figure 4a.
Response R2: We thank the reader for drawing our attention to these values. They have been checked and corrected.
Line 281: Results indicated the teat exert quite different community structure across the time and group, and the same effect was not observed in milk samples. Again there were several fungi shared between teat and bedding material, but not found in milk? While two bacteria shared between teat and milk but not in bedding material. Is there any reason why shared bacteria can be found between teat and milk but not in fungi?
Response R2: There is no biological reasons to have the same proportions of shared bacteria and fungi between teats and milks. They are 2 different kingdoms presenting highly diverse metabolic capacities, various resistance to stress, different colonization skills which also depend on the environments considered…. However, both for fungi and bacteria, milks shared most ASVs with teats (above 70% whatever the condition).
Line 582: Please remove positively in LY because they were not significant as indicated by author.
Response R2: The sentence in Line 362 has been modified has suggested.
